# Unsupervised Network Embedding Beyond Homophily

**Zhiqiang Zhong**                                                    *zhiqiang.zhong@uni.lu*
*University of Luxembourg*

**Guadalupe Gonzalez**                                               *ggg17@ic.ac.uk*
*Imperial College London*

**Daniele Grattarola**                                               *grattd@usi.ch*
*Università della Svizzera italiana*

**Jun Pang**                                                         *jun.pang@uni.lu*
*University of Luxembourg*

**Reviewed on OpenReview:** *https://openreview.net/pdf?id=sRgvmXjrmg*

## Abstract

Network embedding (NE) approaches have emerged as a predominant technique to represent complex networks and have benefited numerous tasks. However, most NE approaches rely on a homophily assumption to learn embeddings with the guidance of supervisory signals, leaving the *unsupervised heterophilous* scenario relatively unexplored. This problem becomes especially relevant in fields where a scarcity of labels exists. Here, we formulate the unsupervised NE task as an $r$-ego network discrimination problem and develop the SELENE framework for learning on networks with homophily and heterophily. Specifically, we design a dual-channel feature embedding pipeline to discriminate $r$-ego networks using node attributes and structural information separately. We employ heterophily adapted self-supervised learning objective functions to optimise the framework to learn intrinsic node embeddings. We show that SELENE's components improve the quality of node embeddings, facilitating the discrimination of connected heterophilous nodes. Comprehensive empirical evaluations on both synthetic and real-world datasets with varying homophily ratios validate the effectiveness of SELENE in homophilous and heterophilous settings showing an up to 12.52% clustering accuracy gain.

## 1 Introduction

Network embedding (NE) has become a predominant approach to finding effective data representations of complex systems that take the form of networks (Cui et al., 2019). NE approaches leveraging graph neural networks (GNNs) (Defferrard et al., 2016; Kipf & Welling, 2017; Xu et al., 2019b; Zhong et al., 2022b) have proven effective in (semi)-supervised settings and achieved remarkable success in various application areas, such as social, e-commerce, biology, and traffic networks (Zhang et al., 2020). However, recent works demonstrated that classic supervised NE methods powered by GNNs, which typically follow a homophily assumption, have limited representation power on heterophilous networks (Bo et al., 2021; Lim et al., 2021; Chien et al., 2021; Zhong et al., 2022a; Zheng et al., 2022).

Whereas similar nodes are connected in homophilous networks, the opposite holds for heterophilous networks in which connected nodes are likely from different classes (Figure 1-(a-b)). For instance, people tend to connect with people of the opposite gender in dating networks (Zhu et al., 2020) and fraudsters are more likely to connect with customers than other fraudsters in online transaction networks (Pandit et al., 2007).

Traditional GNNs typically fail in heterophilous scenarios because they obtain representations by aggregating information from neighbours, acting as a low-frequency filter which generates indistinguishable node

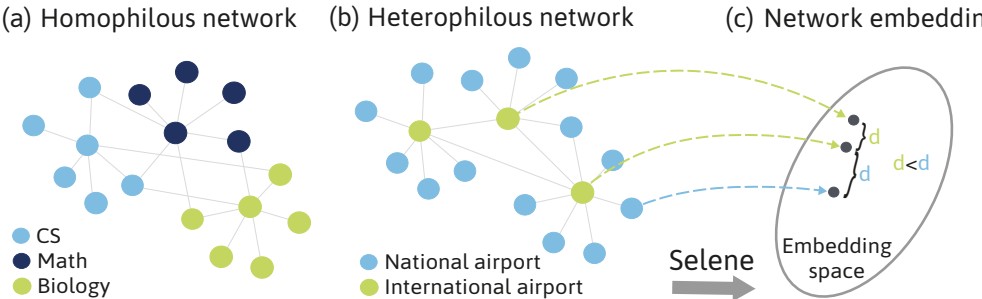

Figure 1: Example homophilous and heterophilous networks ((a): a citation network; (b): an airline transport network). (c) SELENE performs unsupervised network embedding capturing node attributes and structural information to address heterophily.

representations on heterophilous networks (Figure 2) (Bo et al., 2021). Recently, several GNN operators have been introduced to overcome the smoothing effect of traditional GNNs on heterophilous networks (Zheng et al., 2022), however, they rely heavily on the (semi)-supervised setting. Wei et al. (2022) conduct theoretical analyses to find out the conditions that GNN models have no performance difference on synthetic homophilous/heterophilous datasets. Ma et al. (2022) empirically characterise different heterophily conditions and identify the specific conditions that lead to worse GNN model performance under supervised settings. In contrast, the effectiveness of GNNs in unsupervised settings, i.e. learning effective heterophilous representations without any supervision, is relatively unexplored (Xia et al., 2021). In application areas such as biomedical problems, where scarcity of labels exists, the unsupervised setting is of high interest to generate representations for various downstream tasks (Li et al., 2022).

**Current limitations.** We address the task of node clustering under an unsupervised heterophilous setting (Figure 1-(b-c)) (for convenience, we will refer to unsupervised network embedding as NE in the remainder of the text). In this scenario, the first unexplored question to investigate is **RQ1**: *how do existing NE methods perform on heterophilous networks without supervision?*

We conduct an empirical study on 10 synthetic networks with a variety of homophily ratios ($h$, Definition 3) to investigate whether $h$ influences the node clustering performance of representative NE methods. Our experimental results, summarised in Figure 3, show that *(i)* the performance of NE methods that utilise network structure, including heterophilous GNNs designed for supervised settings, decreases significantly when $h \to 0$; *(ii)* the performance of a NE method that only relies on raw node attributes is not affected by changes of $h$, but it is outperformed by other NE methods when $h \to 1$. These two findings directly answer RQ1 and meanwhile raise another interesting and challenging question **RQ2**: *could we design a NE framework that adapts well to both homophily and heterophily settings under no supervision?*

**Our approach.** Motivated by the limitations mentioned above, we approach the NE task as an $r$-ego network discrimination problem and propose the SELf-supErvised Network Embedding (SELENE) framework. Our empirical study suggests that both node attributes and local structure should be leveraged to obtain node embeddings. Therefore, we summarise them into an $r$-ego network and use self-supervised learning (SSL) objective functions to optimise the framework to compute distinguishable node embeddings. Such a design assumes nodes of the same class label share either similar node attributes or $r$-ego network structure. Specifically, we propose a dual-channel embedding pipeline that encodes node attributes and network structure information in parallel (Figure 4). Next, after revisiting representative NE mechanisms, we introduce identity, and network structure features to enhance the framework's ability to capture structural information and distinguish different sampled $r$-ego networks. Lastly, since network sampling strategies often implicitly follow homophily assumptions, i.e., "positively sampling nearby nodes and negatively sampling the faraway nodes" (Yang et al., 2020), we employ negative-sample-free SSL objective functions, namely, reconstruction loss and Barlow-Twins loss, to optimise the framework.

**Extensive evaluation.** We empirically evaluate our model and competitive NE methods on both synthetic and real networks covering the full spectrum of low-to-high homophily and various topics, including node clustering, node classification and link prediction. We observe that SELENE achieves significant performance

gains in both homophily and heterophily in real networks, with an up to 12.52% clustering accuracy gain. Our detailed ablation study confirms the effectiveness of each design component. In synthetic networks $h \in [0, 1)$, we observe that SELENE shows better generalisation.

## 2  Related Work

Node clustering, one of the most fundamental graph analysis tasks, is to group similar nodes into the same category *without supervision* (Ng et al., 2001; Schaeffer, 2007). Over the past decades, many clustering algorithms have been developed and successfully applied to various real-world applications (Belkin & Niyogi, 2001). Recently, the breakthroughs in deep learning have led to a paradigm shift in the machine learning community, achieving great success on many important tasks, including node clustering. Therefore, deep node clustering has caught significant attention (Pouyanfar et al., 2019; Yang et al., 2017). The basic idea of deep node clustering is to integrate the objective of clustering into the powerful representation ability of deep learning. Hence learning an effective node representation is a prerequisite for deep node clustering. To date, network embedding (NE)-based node clustering methods have achieved state-of-the-art performance and become the *de facto* clustering methods.

**Network embedding before GNNs.** NE techniques aim at embedding the node attributes and structure of complex networks into low-dimensional node representations (Cui et al., 2019). Initially, NE was posed as the optimisation of an embedding lookup table directly encoding each node as a vector. Within this group, several methods based on skip-grams (Mikolov et al., 2013) have been proposed, such as DeepWalk (Perozzi et al., 2014), node2vec (Grover & Leskovec, 2016), struc2vec (Ribeiro et al., 2017), etc (Tang et al., 2015; Qiu et al., 2018). Despite the relative success of these NE methods, they often ignore the richness of node attributes and only focus on the network structural information, which hugely limits their performance.

**Network embedding with GNNs.** Recently, GNNs have shown promising results in modelling structural and relational data (Wu et al., 2021). GNN models capture the structural similarity of nodes through a recursive message-passing scheme, using neural networks to implement the message, aggregation, and update functions (Battaglia et al., 2018). The effectiveness of GNNs has been widely proven in (semi)-supervised settings, and they have achieved remarkable success in various areas (Zhang et al., 2020). Several approaches such as ChebNet (Defferrard et al., 2016), GCN (Kipf & Welling, 2017), GraphSAGE (Hamilton et al., 2017), CayleyNets (Levie et al., 2017), GWNN (Xu et al., 2019a), and GIN (Xu et al., 2019b) have led to remarkable breakthroughs in numerous fields in (semi-)supervised settings. However, their effectiveness for unsupervised NE is relatively unexplored. Recently, GNN-based methods for unsupervised NE such as DGI (Velickovic et al., 2019), GMI (Peng et al., 2020), SDCN (Bo et al., 2020), and GBT (Bielak et al., 2021) have been proposed, although they were primarily designed for homophilous networks.

**Heterophilous network embedding.** Recent works have also focused on NE for heterophilous networks and have shown that the representation power of GNNs designed for (semi-)supervised settings is greatly limited on heterophilous networks (Zheng et al., 2022). Some efforts have been dedicated to generalising GNNs to heterophilous networks by introducing complex operations, such as coarsely aggregating higher-order interactions or combining the intermediate representations (Zhu et al., 2020; Bo et al., 2021; Lim et al., 2021). Wei et al. (2022) conduct theoretical analyses to determine the conditions that GNN models have no performance difference on synthetic homophilous/heterophilous datasets. Ma et al. (2022) empirically characterise different heterophily conditions and identify the specific conditions that lead to worse GNN model performance under supervised settings. Nevertheless, these heterophilous GNNs heavily rely on supervisory information and hence cannot be applied to unsupervised settings (verified in Section 4 and Section 6.3). Latterly, Tang et al. (2022) introduce Neighborhood Wasserstein Reconstruction loss and a novel decoder to properly capture node attributes and graph structure to learn discriminate node representations.

## 3  Notation and Preliminaries

An unweighted network can be formally represented as $\mathcal{G} = (\mathcal{V}, \mathcal{E}, \mathbf{X})$, where $\mathcal{V}$ is the set of nodes and $|\mathcal{V}| = n$ is the number of nodes, $\mathcal{E} \subseteq \mathcal{V} \times \mathcal{V}$ is the set of edges, and $\mathbf{X} \in \mathbb{R}^{n \times \pi}$ represents the $\pi$-dimensional node

attributes. We let $\mathcal{Y} = \{y_v\}$ be a set of class labels for all $v \in \mathcal{V}$. For simplicity, we summarise $\mathcal{E}$ with an adjacency matrix $\mathbf{A} \in \{0,1\}^{n \times n}$.

**Problem setup**. We setup the problem using the unsupervised node clustering task as an example. We first learn node representations $\mathbf{Z}_v \in \mathbb{R}^d$ for all $v \in \mathcal{V}$, capturing both node attributes and local network structure. Then, the goal is to infer the unknown class labels $y_v$ for all $v \in \mathcal{V}$ using a clustering algorithm on the learned node representation $\mathbf{Z}_v$ (in this paper, we use the well-known $K$-means algorithm (Hartigan & Wong, 1979)). Note that, for convenience, in the following we refer to unsupervised NE simply as NE.

**Definition 1 (r-hop Neighbourhood $\mathcal{N}^r$)** *We denote the r-hop neighbourhood of node $v$ by $\mathcal{N}_v^r = \{v : d(u,v) \leq r\}$, where $d(u,v)$ is the shortest path distance between $u$ and $v$. For the network shown in Figure 2-(a), $\mathcal{N}_{v_0}^1 = \{v_0, v_1, v_2, v_4, v_7\}$.*

**Definition 2 (r-ego Network) $\mathcal{G}_r(v)$)** *(McAuley & Leskovec, 2012; Qiu et al., 2020) Let $\mathcal{N}_v^r \subseteq \mathcal{V}$ be the r-ego neighbours of node $v$ in $\mathcal{G}$. Their corresponding r-ego network is an induced sub-network of $\mathcal{G}$ defined as $\mathcal{G}_r(v) = \{\mathcal{N}_v^r, \mathcal{E}_v^r, \mathbf{X}_v^r\}$, where $\mathcal{E}_v^r := ((\mathcal{N}_v^r \times \mathcal{N}_v^r) \cap \mathcal{E})$.*

**Definition 3 (Homophily Ratio $h$)** *The homophily ratio $h$ of $\mathcal{G}$ describes the relation between node labels and network structure. Recent works commonly use two measures of homophily, edge homophily ($h_{edge}$) (Zhu et al., 2021) and node homophily ($h_{node}$) (Pei et al., 2020), which can be formulated as $h_{edge} = \frac{|\{(u,v):(u,v)\in\mathcal{E}\wedge y_u=y_v\}|}{|\mathcal{E}|}$, $h_{node} = \frac{1}{|\mathcal{V}|}\sum_{v\in\mathcal{V}}\frac{|\{u:u\in\mathcal{N}_v^1\wedge y_u=y_v\}|}{|\mathcal{N}_v^1|}$ Specifically, where $h_{edge}$ evaluates the fraction of edges between nodes with the same class labels; $h_{node}$ evaluates the overall fraction of neighbouring nodes that have the same class labels. In this paper, we focus only on edge homophily and denote it with $h = h_{edge}$. Figure 2-(a) shows an example network with $h = 0.2$.*

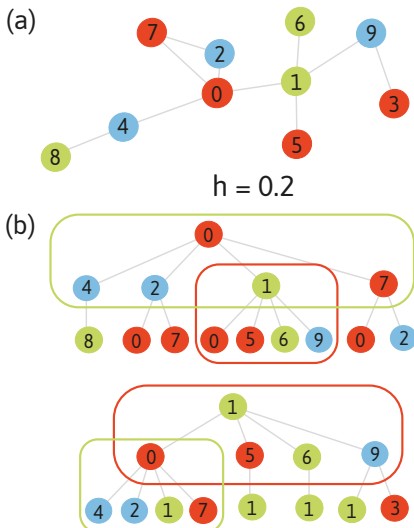

Figure 2: (a) Network with $h = 0.2$; (b) the aggregation mechanism in traditional GNNs implicitly follows a homophily assumption as a result of duplicated aggregation trees for nearby nodes - as shown for trees rooted in nodes $v_0$ and $v_1$ (duplicates marked by same-coloured square).

## 4 An Experimental Investigation

In this section, we empirically analyse the performance of NE methods on 10 synthetic networks with different homophily ratios ($h$). The main goal is to investigate (**RQ1**): *how do existing NE methods perform on heterophilous networks?* Specifically, we quantify their performance on the node clustering task on 10 synthetic networks with $h \in [0, 0.1, \ldots, 0.9]$. A detailed description of the synthetic datasets generation process can be found in Section 6.1, and we refer the reader to Section 6.2 for details on the experimental settings.

Figure 3 illustrates that the clustering accuracy of representative NE methods that utilise network structure, i.e., node2vec (Grover & Leskovec, 2016), GAE (Kipf & Welling, 2016), GraphSAGE (Hamilton et al., 2017), SDCN (Bo et al., 2020), GBT (Bielak et al., 2021) and H2GCN* (Zhu et al., 2020) show outstanding performance when $h \to 1$ but with the decrease of $h$, their performance decreases significantly. The reason why existing NE methods using network structure fail

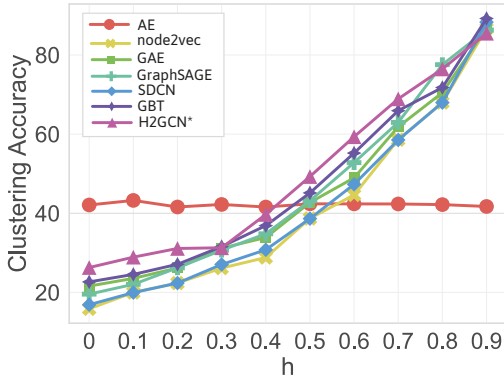

Figure 3: Node clustering accuracy of representative NE methods on synthetic networks.

only when $h \to 0$ is that most of them implicitly follow a homophily assumption, with specific objective function or aggregation mechanism designs. For instance, *(i)* objective functions of node2vec and Graph-

SAGE guide nodes at close distance to have similar representations and nodes far away to have different ones; *(ii)* the inherent aggregation mechanism of GAE, GraphSAGE, SDCN and GBT naturally assumes local smoothing (Chen et al., 2020) (which is mainly caused by the duplicated aggregation tree for nearby nodes as shown in Figure 2-(b)), which translates into neighbouring nodes having similar representations.

On the other hand, H2GCN*, which is specifically designed for heterophilous networks, follows the same trend as the homophilous approaches due to the loss of supervisory signals in the network embedding task. The only method with stable performance across different values of $h$ is AE (Hinton & Salakhutdinov, 2006), which is attributable to its reliance on raw node attributes only. AE exhibits an apparent advantage against all other models when $h < 0.5$. This highlights the importance of considering node attributes in the design of NE approaches for networks with heterophily.

# 5  Network Embedding via $r$-ego Network Discrimination

In this section, we formalise the main challenges of NE on heterophilous networks. To address these challenges, we present the $\underline{\text{SEL}}$f-sup$\underline{\text{E}}$rvised $\underline{\text{N}}$etwork $\underline{\text{E}}$mbedding (SELENE) framework. Figure 4 shows the overall view of SELENE.

**Challenges**. Motivated by the empirical results in Section 4, we realise that a NE method for heterophilous networks should have the ability to distinguish nodes with different attributes or structural information. Each node's local structure can be flexibly defined by its $r$-ego network. We further summarise each node's relevant node attribute and structural information into an $r$-ego network and define the NE task as an $r$-ego network discrimination problem. Then, we address three main research challenges to solve this problem: (**RC1**) How to leverage node attributes and network structure for NE? (**RC2**) How to break the inherent homophily assumptions of traditional NE mechanisms? (**RC3**) How to define an appropriate objective function to optimise the embedding learning process?

The following subsections discuss our solutions to address these three challenges and introduce the SELENE framework. In Section 6, we provide a comprehensive empirical evaluation on both synthetic and real data with varying homophily ratios to validate the effectiveness of SELENE under homophilous and heterophilous settings. We also show that all components in our design are helpful in improving the quality of node embeddings.

## 5.1  Dual-channel Feature Embedding (RC1)

Motivated by the empirical observation in Section 4 that AE (only based on raw node attributes) has the only stable performance across $h \in [0, 1)$ and graph structure-based NE approaches show priority when $h \to 1$. Therefore, to learn node embeddings that can discriminate $r$-ego networks of different nodes, we propose to deal with two important perspectives, i.e., node attributes and graph structure, separately. We first extract the $r$-ego network $(\mathcal{G}_r(v))$ of each node $v \in \mathcal{G}$. For example, in Figure 4, $\mathcal{G}_2(v)$ represents a 2-ego network instance of $\mathcal{G}$. The empirical analysis of Section 4 highlighted that node attributes and structural information play a major role in discriminating nodes over networks. Therefore, we propose a dual-channel feature embedding pipeline to learn node representation from node attributes and network structure separately, as shown in Figure 4. That said, we split $r$-ego network of each node $\mathcal{G}_r(v) = \{\mathcal{N}_v^r, \mathcal{E}_v^r, \mathbf{X}_v^r\}$ into ego node attribute $\mathbf{X}_v$ and network structure $\widetilde{\mathcal{G}}_r(v) = \{\mathcal{N}_v^r, \mathcal{E}_v^r\}$. And in order to avoid dense ego networks, we adopt one widely adopted neighbour sampler (Hamilton et al., 2017) to restrict the number of sampled nodes of each hop is no more than 15. Appendix C will provide more description about the implementation. Such a design brings two main benefits: *(i)* both important sources of information can be well utilised without interfering with each other, and *(ii)* the inherent homophily assumptions of NE methods can be greatly alleviated, an issue we will address in the following subsection. We will empirically discuss the effectiveness of the dual channel feature embedding pipeline in Section 6.4.

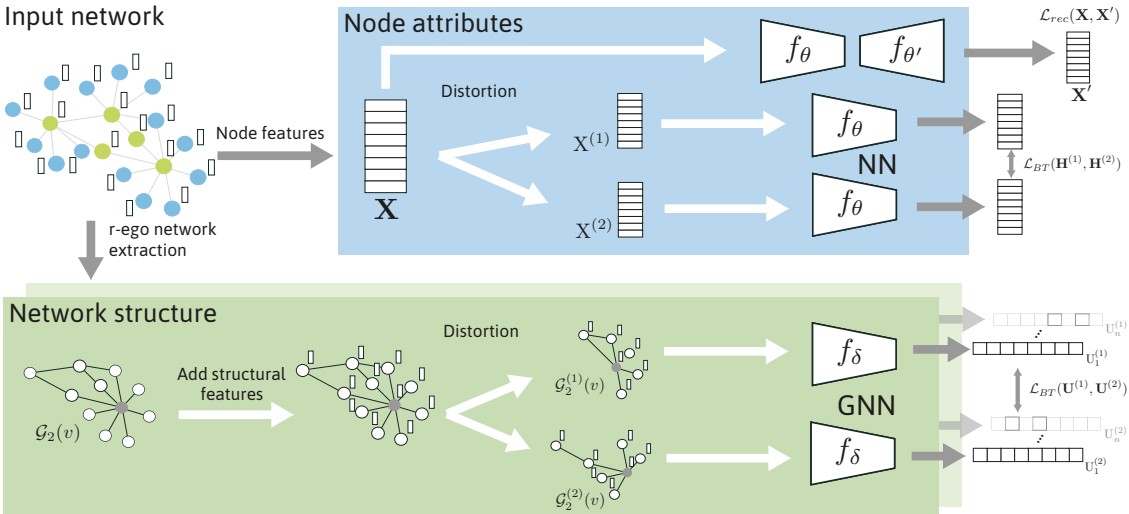

Figure 4: We propose a dual-channel feature embedding pipeline to learn node representations separately from node attributes and network structure. Node feature attribute embeddings are optimised using an autoencoder trained to reconstruct the input data $\mathcal{L}_{Rec}(\mathbf{X}, \widehat{\mathbf{X}})$, and refined through the contrastive loss $\mathcal{L}_{BT}(\mathbf{H}^{(1)}, \mathbf{H}^{(2)})$. The network structure encoder is optimised using $\mathcal{L}_{BT}(\mathbf{U}^{(1)}, \mathbf{U}^{(2)})$. The final node representation is obtained by applying COMBINE to the generated node attribute representation $\mathbf{H}$ and network structure representation $\mathbf{U}$.

## 5.2 *r*-ego Network Feature Extraction (RC2)

**Node attribute encoder module.** As previously mentioned, learning effective node attribute representations is of great importance for NE. In this paper, we employ the basic Autoencoder (Hinton & Salakhutdinov, 2006) to learn representations of raw node attributes, which can be replaced by more sophisticated encoders (Masci et al., 2011; Makhzani et al., 2015; Malhotra et al., 2016) to obtain higher performance. We assume an $L$-layers Autoencoder ($f_\theta$), with the formulation of the $\ell$-th encoding layer being:

$$\mathbf{H}_e^{(\ell)} = \phi(\mathbf{W}_e^{(\ell)}\mathbf{H}_e^{(\ell-1)} + \mathbf{b}_e^{(\ell)}) \tag{1}$$

where $\phi$ is a non-linear activation function such as ReLU (Nair & Hinton, 2010) or PReLU (He et al., 2015). $\mathbf{H}_e^{(\ell-1)} \in \mathbb{R}^{n \times d_{\ell-1}}$ is the hidden node attribute representations in layer $\ell - 1$, with $d_{\ell-1}$ being the dimensionality of this layer's hidden representation. $\mathbf{W}_e^{(\ell)} \in \mathbb{R}^{d_{\ell-1} \times d_\ell}$ and $\mathbf{b}_e^{(\ell)} \in \mathbb{R}^{d_\ell}$ are trainable weight matrix and bias of the $\ell$-th layer in the encoder. Node representations $\mathbf{H}_v = f_\theta(\mathbf{X}_v) = \mathbf{H}_e^{(L)}$ are obtained after successive application of $L$ encoding layers.

**Identity and network structure features.** Despite the significant success of GNNs in a variety of network-related tasks, their representation power in network structural representation learning is limited (Xu et al., 2019b). In order to obtain invariant node structural representations so that nodes with different ego network structures are assigned different representations, we employ the identity feature (You et al., 2021) and structural features (Li et al., 2020). In this paper, we adopt variant shortest path distance (SPD) as node structural features ($\widetilde{\mathbf{X}}_{struc}$). After, we further inject the node identity features ($\widetilde{\mathbf{X}}_{id}$) as augment features, hence we have node feature matrix $\widetilde{\mathbf{X}} = \widetilde{\mathbf{X}}_{struc} + \widetilde{\mathbf{X}}_{id}$ for each ego-network.

**Network structure encoder module.** Over the past few years, numerous GNNs have been proposed to learn node representations from network-structured data, including spectral GNNs (i.e., ChebNet (Defferrard et al., 2016), CayletNet (Levie et al., 2017) and GWNN (Xu et al., 2019a)) and spatial GNNs (i.e., GraphSAGE (Hamilton et al., 2017), GAT (Velickovic et al., 2018), GIN (Xu et al., 2019b)). For the sake of simplicity, we adopt a simple GNN variant, i.e., GCN (Kipf & Welling, 2017), as the building block of the

network structure encoder ($f_\delta$). The $\ell$-th layer of a GCN for $v$'s $r$-ego network can be formally defined as:

$$\mathbf{U}^{(\ell)} = \sigma(\widehat{\mathbf{D}}^{-\frac{1}{2}}\widehat{\mathbf{A}}\widehat{\mathbf{D}}^{-\frac{1}{2}}\mathbf{U}^{(\ell-1)}\mathbf{W}^{(\ell)}) \tag{2}$$

with $\widehat{\mathbf{A}} = \mathbf{A} + \mathbf{I}$, where $\mathbf{I}$ is the identity matrix, and $\widehat{\mathbf{D}}$ is the diagonal node degree matrix of $\widehat{\mathbf{A}}$. $\mathbf{U}^{(\ell-1)} \in \mathbb{R}^{n \times d_{\ell-1}}$ is the hidden representation of nodes in layer $\ell - 1$, with $d_{\ell-1}$ being the dimensionality of this layer's representation, and $\mathbf{U}^0 = \widetilde{\mathbf{X}}$. $\mathbf{W}^{(\ell)} \in \mathbb{R}^{d_{\ell-1} \times d_\ell}$ is a trainable parameter matrix. $\sigma$ is a non-linear activation function such as ReLU or Sigmoid (Han & Moraga, 1995) function. Structural representations $\mathbf{U}_v = f_\delta(\widetilde{\mathcal{G}}_r(v)) = \mathbf{U}^{(L)}$ are obtained after successive applications of $L$ layers.

## 5.3 Heterophily Adapted Self-Supervised Learning (RC3)

The objective function plays a significant role in NE tasks. Several objective functions have been proposed for NE, such as network reconstruction loss (Kipf & Welling, 2016), distribution approximating loss (Perozzi et al., 2014) and node distance approximating loss (Hamilton et al., 2017). Nevertheless, none of them is suitable for NE optimisation on heterophilous networks because of the homophily assumptions used to determine (dis)similar pairs. In heterophilous networks, node distance on network alone does not determine (dis)similarity, i.e., connected nodes are not necessarily similar, and nodes far apart are not necessarily dissimilar. This removes the need for connected nodes to be close in the embedding space and for disconnected nodes to be far apart in the embedding space.

Inspired by the latest success of negative-sample-free self-supervised learning (SSL), we adopt the Barlow-Twins (BT) (Zbontar et al., 2021) as our overall optimisation objective. Consequently, the invariance term makes the representation invariant to the distortions applied; the redundancy reduction term lets the representation units contain non-redundant information about the target sample. More details about the BT method can be found in Appendix E. We discuss the feasibility of different objective functions in Section 6.4.

**Distortion.** To apply the BT method to optimise our framework, we have to first generate distorted samples for target $r$-ego network instances. Due to the dual-channel feature embedding pipeline, we introduce $f_{Aug}^{\mathbf{X}}$ and $f_{Aug}^{\mathcal{G}}$ to generate distorted node attribute and network structure instances, respectively. Our data distortion method adopts a similar strategy as You et al. (2020); Bielak et al. (2021) that randomly masks node attributes and edges with probability $p_x, p_e$. Formally, this distortion process can be represented as:

$$f_{Aug}^{\mathbf{X}}(\mathbf{X}_v, p_x) = (\mathbf{X}_v^{(1)}, \mathbf{X}_v^{(2)})$$
$$f_{Aug}^{\mathcal{G}}(\widetilde{\mathcal{G}}_r(v), p_x, p_e) = (\widetilde{\mathcal{G}}_r^{(1)}(v), \widetilde{\mathcal{G}}_r^{(2)}(v)) \tag{3}$$

**Barlow-Twins loss function.** Based on the two pairs of distorted instances, two pairs of representations $(\mathbf{U}_v^{(1)}, \mathbf{U}_v^{(2)})$ and $(\mathbf{H}_v^{(1)}, \mathbf{H}_v^{(2)})$ can be computed by applying $f_\theta$ and $f_\delta$, respectively. Then, BT method can be employed to evaluate the computed representations and guide the framework optimisation. Using representations $(\mathbf{H}^{(1)}, \mathbf{H}^{(2)})$ as example, the loss value is formally computed as:

$$\mathcal{L}_{BT}(\mathbf{H}^{(1)}, \mathbf{H}^{(2)}) = \sum_i^{|\mathcal{V}|}(1 - \mathcal{C}_{ii})^2 + \lambda \sum_i^{|\mathcal{V}|} \sum_{j \neq i}^{|\mathcal{V}|} \mathcal{C}_{ij}^2 \tag{4}$$

where $\mathcal{C}_{ij} = \frac{\sum_b \mathbf{H}_{b,i}^{(1)}\mathbf{H}_{b,j}^{(2)}}{\sqrt{\sum_b (\mathbf{H}_{b,j}^{(1)})^2}\sqrt{\sum_b (\mathbf{H}_{b,j}^{(2)})^2}}$. $\lambda > 0$ defines the trade-off between the invariance and redundancy reduction terms, $b$ is the batch indexes, and $i$, $j$ index the vector dimension of the input representation vectors. We adopt the default settings as Zbontar et al. (2021).

**Node attribute reconstruction loss function.** In addition to the general objective function, $\mathcal{L}_{BT}$, we adopt another objective function, i.e., node attribute reconstruction loss $\mathcal{L}_{Rec}$, to optimise the node attribute encoder specifically. Following the node attribute encoder (Eq.1), the decoder reconstructs input node attributes from the computed node representations $\mathbf{H}$. Typically, a decoder has the same structure as the encoder by reversing the order of layers. Its $\ell$-th fully connected layer can be formally represented:

$\mathbf{H}_d^{(\ell)} = \phi(\mathbf{W}_d^{(\ell)}\mathbf{H}_d^{(\ell-1)} + \mathbf{b}_d^{(\ell)})$. Reconstructed node attributes $\widehat{\mathbf{X}} = \mathbf{H}_d^{(L)}$ are obtained after successive applications of $L$ decoding layers. We optimise the autoencoder parameters by minimising the difference between raw node attributes $\mathbf{X}$ and reconstructed node attributes $\widehat{\mathbf{X}}$ with:

$$\mathcal{L}_{Rec}(\mathbf{X}, \widehat{\mathbf{X}}) = \frac{1}{2|\mathcal{V}|}||\mathbf{X} - \widehat{\mathbf{X}}||_F^2 \tag{5}$$

Empowered with Barlow-Twins and node attribute reconstruction loss, we can optimise the framework's encoders under heterophilous settings. The node attribute encoder is optimised with $\mathcal{L}_{BT}$ (Eq. 4) and $\mathcal{L}_{Rec}$ (Eq. 5), and the network structure encoder is optimised with $\mathcal{L}_{BT}$ (Eq. 4). The overall loss function is $\mathcal{L} = \mathcal{L}_{BT}(\mathbf{U}^{(1)}, \mathbf{U}^{(2)}) + \mathcal{L}_{BT}(\mathbf{H}^{(1)}, \mathbf{H}^{(2)}) + \mathcal{L}_{Rec}(\mathbf{X}, \widehat{\mathbf{X}})$.

**Final node representations.** Representations capturing node attributes ($\mathbf{H}$) and structural context ($\mathbf{U}$) are combined to obtain expressive and powerful representations as: $\mathbf{Z} = \text{COMBINE}(\mathbf{H}, \mathbf{U})$ where COMBINE($\cdot$) can be any commonly used operation in GNNs (Xu et al., 2019b), such as *mean*, *max*, *sum* and *concat*. We utilise *concat* in all our experiments, allowing for an independent integration of representations learnt by the dual-channel architecture.

### 5.4 Summary

---

**Algorithm 1:** SELf-supErvised Network Embedding (SELENE) Framework

---

**Input:** Network $\mathcal{G} = (\mathcal{V}, \mathcal{E}, \mathbf{X})$ ;
Node attribute encoder $f_\theta$ and decoder $f_{\theta'}$ ;
Network structure encoder (GNN) $f_\delta$ ;
Node attribute distortion function $f_{Aug}^{\mathbf{x}}$ and network distortion function $f_{Aug}^{\mathcal{G}}$ ;
**Output:** node representations $\mathbf{Z}$

1   Sample a set of $r$-ego networks $\{\mathcal{G}_r(1), \mathcal{G}_r(2), \ldots, \mathcal{G}_r(n)\}$ from $\mathcal{G}$ ;

2   Extract network structure of each $r$-ego network $\mathcal{G}_r(v)$ and enhance each with identity features ($\widetilde{\mathbf{X}}_{id}$) and network structure features ($\widetilde{\mathbf{X}}_{struc}$) to obtain $\widetilde{\mathcal{G}}_r(v) = \{\mathcal{N}_v^r, \mathcal{E}_v^r, \widetilde{\mathbf{X}}\}$ ;

3   Generate distorted node attribute matrix instances $\mathbf{X}^{(1)}$ and $\mathbf{X}^{(2)}$ ;

4   Generate two distorted instances $(\widetilde{\mathcal{G}}_r^{(1)}(v), \widetilde{\mathcal{G}}_r^{(2)}(v))$ for each $r$-ego network $\widetilde{\mathcal{G}}_r(v)$ ;

5   **repeat**

6      Initialise loss $\mathcal{L}$ as zero ;

7      **for** *each node $v \in \mathcal{V}$* **do**

8          $\mathbf{H}_v = f_\theta(\mathbf{X}_v), \qquad \widehat{\mathbf{X}}_v = f_{\theta'}(\mathbf{H}_v)$ ;

9          $\mathbf{H}_v^{(1)} = f_\theta(\mathbf{X}_v^{(1)}), \qquad \mathbf{H}_v^{(2)} = f_\theta(\mathbf{X}_v^{(2)})$ ;

10        $\mathbf{U}_v^{(1)} = f_\delta(\widetilde{\mathcal{G}}_r^{(1)}(v)), \quad \mathbf{U}_v^{(2)} = f_\delta(\widetilde{\mathcal{G}}_r^{(2)}(v))$ ;

11        $\mathcal{L} = \mathcal{L}_{BT}(\mathbf{U}_v^{(1)}, \mathbf{U}_v^{(2)}) + \mathcal{L}_{BT}(\mathbf{H}_v^{(1)}, \mathbf{H}_v^{(2)}) + \mathcal{L}_{Rec}(\mathbf{X}_v, \widehat{\mathbf{X}}_v)$ ;

12      **end**

13      Update $\theta$ and $\delta$ by descending the gradients $\nabla_{\theta,\delta}\mathcal{L}$ ;

14 **until** *Convergence*;

15 $\mathbf{H} = f_\theta(\mathbf{X})$ ;

16 **for** *each node $v \in \mathcal{V}$* **do**

17      $\mathbf{U}_v = f_\delta(\widetilde{\mathcal{G}}_r(v))$ ;

18 **end**

19 $\mathbf{Z} = \text{COMBINE}(\mathbf{H}, \mathbf{U})$ ;

---

Here, we summarise SELENE in Algorithm 1 to provide a general overview of our framework. Given as input a graph $\mathcal{G}$ and node attribute encoder $f_\theta$ and decoder $f_{\theta'}$, network structure encoder $f_\delta$, node attribute distortion function $f_{Aug}^{\mathbf{X}}$, and network distortion function $f_{Aug}^{\mathcal{G}}$. Our algorithm is motivated by the empirical results in Section 4, which showed that a NE method for heterophilous networks should have the ability to

distinguish nodes with different attributes or structural information. Therefore, we summarise each node's relevant node attribute and structural information into an $r$-ego network and define the NE task as an $r$-ego network discrimination problem. We sample a set of $r$-ego networks $\{\mathcal{G}_r(1), \mathcal{G}_r(2), \ldots, \mathcal{G}_r(n)\}$ from $\mathcal{G}$ (Line 1). Next, we extract network structure of each $r$-ego network $\mathcal{G}_r(v)$ and enhance each with identity features ($\widetilde{\mathbf{X}}_{id}$) and network structure features ($\widetilde{\mathbf{X}}_{struc}$) to obtain $\widetilde{\mathcal{G}}_r(v) = \{\mathcal{N}_v^r, \mathcal{E}_v^r, \widetilde{\mathbf{X}}\}$ (Line 2). Then, we generate distorted node attribute matrix instances $\mathbf{X}^{(1)}$ and $\mathbf{X}^{(2)}$ and two distorted instances $(\widetilde{\mathcal{G}}_r^{(1)}(v), \widetilde{\mathcal{G}}_r^{(2)}(v))$ for each $r$-ego network $\widetilde{\mathcal{G}}_r(v)$ (Line 3-4). For each node $v \in \mathcal{G}$, we compute its node attribute embedding ($\mathbf{H}_v$) and distorted node attribute embeddings ($\mathbf{H}_v^{(1)}, \mathbf{H}_v^{(2)}$), and network structure embeddings ($\mathbf{U}_v^{(1)}, \mathbf{U}_v^{(2)}$) (Line 8-10). Then, Barlow-Twins and network attribute reconstruction loss functions are employed to compute the final loss $\mathcal{L}$ (Line 11). We update SELENE's parameters ($\theta, \delta$) by descending the gradient $\nabla_{\theta,\delta}\mathcal{L}$ (Line 13) until convergence. After the model is trained, we compute node attribute embedding $\mathbf{H}_v$ and network structure embedding $\mathbf{U}_v$ to obtain the final node representations $\mathbf{Z}$ by combining $\mathbf{H}$ and $\mathbf{U}$ (Line 15-19).

**Discussion.** The description above indicates that the design of SELENE assumes nodes of the same class label share either similar node attributes or $r$-ego network structure. Under this assumption, we can learn node representations ($\mathbf{Z}$) to distinguish nodes of different class labels from the perspective of node attributes ($\mathbf{H}$) or $r$-ego network structure ($\mathbf{U}$). Nevertheless, if nodes of different class labels have similar node attributes and $r$-ego network structure, it is hard for SELENE to distinguish them. We address this limitation as a very interesting future work to explore.

## 6 Evaluation

### 6.1 Real-world and Synthetic Datasets

**Real-world datasets.** We use a total of 12 real-world datasets (Texas (Pei et al., 2020), Wisconsin (Pei et al., 2020), Actor (Pei et al., 2020), Chameleon (Rozemberczki et al., 2021), USA-Airports (Ribeiro et al., 2017), Cornell (Pei et al., 2020), Europe-Airports (Ribeiro et al., 2017), Brazil-Airports (Ribeiro et al., 2017), Deezer-Europe (Rozemberczki & Sarkar, 2020), Citeseer (Kipf & Welling, 2017), DBLP (Fu et al., 2020), Pubmed (Kipf & Welling, 2017)) in diverse domains (web-page, citation, co-author, flight transport and online user relation). All real-world datasets are available online[1]. Statistics information is summarised in Table 4 of Appendix A.

**Synthetic datasets.** Moreover, we generate random synthetic networks with various homophily ratios $h$ by adopting a similar approach to Abu-El-Haija et al. (2019); Kim & Oh (2021). Specifically, each synthetic network has 10 classes and 500 nodes per class. Nodes are assigned random features sampled from 2D Gaussians, and each dataset has 10 networks with $h \in [0, 0.1, 0.2, \ldots, 0.9]$. The detailed data generation process can be found in Appendix A.

### 6.2 Experimental Setup

We compare our framework SELENE [2] with 12 competing NE methods in terms of the different challenging graph analysis tasks, including node clustering, node class prediction and link prediction. We adopt 2 different competing NE methods without NNs, including node2vec (N2V) (Grover & Leskovec, 2016) and struc2vec (S2V) (Ribeiro et al., 2017). We adopt 10 additional competing NE methods using NNs, including AE (Hinton & Salakhutdinov, 2006), GAE (Kipf & Welling, 2016), GraphSAGE (SAGE) (Hamilton et al., 2017), DGI (Velickovic et al., 2019), SDCN (Bo et al., 2020), GMI (Peng et al., 2020), GBT (Bielak et al., 2021), H2GCN (Zhu et al., 2020), FAGCN (Bo et al., 2021) and GPRGNN (Chien et al., 2021). Note that GBT is an SSL approach that applies the Barlow-Twins (Zbontar et al., 2021) strategy to network-structured data. Albeit providing a new model training strategy, the basic GNN building blocks remain the same, hence still maintaining a homophily assumption. H2GCN, FAGCN and GPRGNN are state-of-the-art (SOTA) heterophilous GNN operators for supervised settings. Here, we train them using the same mechanism as GBT to adapt them to the unsupervised setting. We rename the unsupervised adaptations of

---

[1] https://pytorch-geometric.readthedocs.io/en/latest/modules/datasets.html
[2] Code and data are available at: https://github.com/zhiqiangzhongddu/SELENE

Table 1: Node clustering results on real-world datasets. The **bold** and underline numbers represent the top-2 results. OOM: out-of-memory.

| Dataset | Metrics | AE | N2V | S2V | GAE | SAGE | SDCN | DGI | GMI | GBT | H2GCN* | FAGCN* | GPRGNN* | Ours | ↑ (%) |
|---|---|---|---|---|---|---|---|---|---|---|---|---|---|---|---|
| | | | | | | | Heterophilous datasets | | | | | | | | |
| Texas $h = 0.11$ | ACC | 50.49 | 48.80 | 49.73 | 42.02 | 56.83 | 44.04 | 55.74 | 35.19 | 55.46 | 58.80 | 57.92 | 57.50 | **65.23** | 10.94 |
| | NMI | 16.63 | 2.58 | 18.61 | 8.49 | 16.97 | 14.24 | 8.73 | 7.72 | 10.17 | 22.49 | 23.35 | 22.83 | **25.40** | 8.78 |
| | ARI | 14.6 | -1.62 | 20.97 | 10.83 | 23.50 | 10.65 | 8.25 | 2.96 | 12.10 | 25.04 | 22.54 | 23.51 | **34.21** | 36.62 |
| Wisc. $h = 0.20$ | ACC | 58.61 | 41.39 | 43.03 | 37.81 | 46.29 | 38.25 | 44.58 | 36.97 | 48.01 | 64.18 | 61.91 | 63.84 | **71.69** | 11.70 |
| | NMI | 30.92 | 4.23 | 11.23 | 9.19 | 10.16 | 8.46 | 10.72 | 11.68 | 7.55 | 29.64 | 27.35 | 29.54 | **39.51** | 27.78 |
| | ARI | 28.53 | -0.48 | 11.50 | 5.2 | 6.06 | 3.67 | 10.31 | 3.74 | 3.85 | 32.61 | 31.56 | 30.53 | **43.48** | 33.33 |
| Actor $h = 0.22$ | ACC | 24.19 | 25.02 | 22.49 | 23.45 | 23.08 | 23.67 | 24.26 | 26.18 | 24.68 | 25.55 | 25.61 | 25.80 | **29.03** | 12.52 |
| | NMI | 0.97 | 0.09 | 0.04 | 0.18 | 0.58 | 0.08 | 1.38 | 0.20 | 0.74 | 3.23 | 3.22 | 3.21 | **4.72** | 46.13 |
| | ARI | 0.50 | 0.06 | -0.05 | -0.04 | 0.22 | -0.01 | 0.07 | 0.41 | -0.57 | 0.31 | 0.34 | 0.31 | **1.84** | 268.00 |
| Chamel. $h = 0.23$ | ACC | 35.68 | 21.31 | 26.34 | 32.76 | 31.04 | 33.5 | 27.77 | 25.73 | 32.21 | 30.62 | 31.33 | 34.62 | **38.97** | 9.22 |
| | NMI | 10.38 | 0.34 | 3.55 | 11.60 | 10.55 | 9.57 | 4.42 | 2.5 | 10.56 | 14.62 | 14.71 | 10.31 | **20.63** | 40.24 |
| | ARI | 5.80 | 0.02 | 1.82 | 4.65 | 6.16 | 5.86 | 1.85 | 0.52 | 7.01 | 4.78 | 5.16 | 5.01 | **15.94** | 127.39 |
| USA-Air. $h = 0.25$ | ACC | 55.24 | 26.29 | 27.58 | 30.84 | 32.96 | 33.52 | 33.36 | 28.69 | 34.96 | 39.01 | 38.82 | 39.63 | **58.90** | 6.63 |
| | NMI | 30.13 | 0.25 | 0.44 | 2.71 | 2.67 | 5.21 | 5.52 | 0.6 | 5.27 | 12.43 | 12.30 | 12.02 | **31.17** | 3.45 |
| | ARI | 24.20 | -0.05 | 0.09 | 2.67 | 2.52 | 1.93 | 4.95 | 0.29 | 3.42 | 9.48 | 9.33 | 9.02 | **25.53** | 5.49 |
| Cornell $h = 0.31$ | ACC | 52.19 | 50.98 | 32.68 | 43.72 | 44.7 | 36.94 | 44.1 | 33.55 | 52.19 | 54.97 | 56.23 | 55.33 | **57.96** | 3.08 |
| | NMI | 17.08 | 5.84 | 1.54 | 5.11 | 4.33 | 6.6 | 5.79 | 5.26 | 5.94 | 17.05 | 17.08 | 16.90 | **17.32** | 1.41 |
| | ARI | 17.41 | 0.18 | -2.20 | 6.51 | 5.64 | 3.58 | 4.87 | 3.05 | 3.05 | 19.50 | 19.88 | 19.21 | **23.03** | 15.85 |
| Eu.-Air. $h = 0.31$ | ACC | 55.36 | 30.78 | 36.89 | 34.84 | 31.75 | 37.37 | 35.59 | 35.34 | 39.75 | 37.27 | 42.11 | 36.63 | **57.80** | 4.41 |
| | NMI | 32.44 | 3.69 | 6.15 | 10.15 | 2.10 | 8.45 | 10.77 | 11.08 | 9.44 | 9.08 | 16.81 | 11.60 | **34.25** | 5.58 |
| | ARI | 24.24 | 0.83 | 4.49 | 7.37 | 1.16 | 5.31 | 8.44 | 8.18 | 7.87 | 5.3 | 11.98 | 9.41 | **25.69** | 5.98 |
| Bra.-Air. $h = 0.31$ | ACC | 71.68 | 30.38 | 38.93 | 36.64 | 37.02 | 38.7 | 37.1 | 38.93 | 40.92 | 43.97 | 44.2 | 46.42 | **79.12** | 10.38 |
| | NMI | 49.26 | 2.5 | 10.23 | 10.96 | 6.89 | 14.05 | 10.64 | 12.62 | 12.16 | 22.37 | 22.67 | 20.10 | **55.90** | 13.48 |
| | ARI | 42.93 | -0.22 | 5.45 | 6.56 | 4.18 | 7.27 | 7.02 | 9.11 | 8.31 | 13.99 | 14.4 | 15.06 | **53.21** | 23.95 |
| Deezer $h = 0.53$ | ACC | 55.88 | 52.97 | OOM | 51.51 | 51.06 | 54.76 | 53.16 | OOM | OOM | 56.81 | 56.81 | 53.22 | **59.94** | 5.51 |
| | NMI | 0.28 | 0.0 | OOM | 0.13 | 0.16 | 0.17 | 0.05 | OOM | OOM | 0.27 | 0.27 | 0.01 | **0.34** | 21.43 |
| | ARI | 0.81 | 0.02 | OOM | 0.07 | -0.02 | 0.61 | -0.23 | OOM | OOM | 1.22 | 0.82 | 0.77 | **1.33** | 9.02 |
| | | | | | | | Homophilous datasets | | | | | | | | |
| Citeseer $h = 0.74$ | ACC | 58.79 | 20.76 | 21.22 | 48.37 | 49.28 | 59.86 | 58.94 | 59.04 | 57.21 | 47.33 | 47.42 | 45.83 | **60.02** | 0.27 |
| | NMI | 30.91 | 0.35 | 1.18 | 24.59 | 22.97 | 30.37 | 32.6 | 32.11 | 31.9 | 20.48 | 20.18 | 20.58 | **32.74** | 0.31 |
| | ARI | 30.29 | -0.01 | 0.17 | 19.50 | 19.21 | 29.7 | 33.16 | 33.09 | 33.17 | 18.03 | 17.93 | 18.11 | **33.46** | 0.87 |
| DBLP $h = 0.80$ | ACC | 48.50 | 29.19 | 31.65 | 57.81 | 48.68 | 61.94 | 58.22 | 63.28 | 73.10 | 41.87 | 41.25 | 43.96 | **75.74** | 3.61 |
| | NMI | 18.98 | 0.14 | 1.33 | 28.94 | 16.46 | 27.13 | 29.98 | 33.91 | 42.21 | 11.04 | 10.60 | 11.27 | **44.17** | 4.64 |
| | ARI | 15.15 | -0.04 | 1.39 | 18.78 | 13.38 | 27.77 | 26.81 | 28.77 | 42.57 | 4.67 | 4.32 | 4.92 | **46.35** | 8.88 |
| Pubmed $h = 0.80$ | ACC | 65.34 | 39.32 | 37.39 | 42.08 | 67.66 | 61.9 | 65.47 | OOM | OOM | 56.71 | 56.88 | 58.81 | **67.98** | 0.47 |
| | NMI | 26.89 | 0.02 | 0.07 | 1.28 | 30.71 | 19.71 | 28.05 | OOM | OOM | 17.15 | 16.91 | 17.01 | **31.14** | 1.40 |
| | ARI | 25.98 | 0.09 | 0.06 | 0.15 | 29.10 | 18.63 | 27.25 | OOM | OOM | 16.51 | 16.27 | 16.24 | **29.79** | 2.37 |

H2GCN, FAGCN and GPRGNN as H2GCN*, FAGCN* and GPRGNN*, respectively. See Appendix B and Appendix C for details of all competing methods and implementation.

## 6.3 Experimental Results

**Node clustering.** Node clustering results on homophilous and heterophilous *real-world datasets* are summarised in Table 1 where we see that SELENE is the best-performing method in all heterophilous datasets. In particular, compared to the best results of competing models, our framework achieves a significant improvement of up to 12.52% on ACC, 46.13% on NMI and 268% on ARI. Such outstanding performance gain demonstrates that SELENE successfully integrates the important node attributes and network structure information into node representations. An interesting case is the comparison of SELENE, H2GCN*, FAGCN* and GBT, given that H2GCN*, FAGCN* and GBT also utilise the Barlow-Twins objective function to optimise a GNN model, and the major difference between H2GCN*, FAGCN*, GBT and SELENE is our designs to address **RC1** and **RC2**. SELENE has a superior performance in all heterophilous datasets, indicating the effectiveness of designs and showing that simply porting supervised models to unsupervised scenarios is not appropriate. Results of node clustering on homophilous networks, contained in Table 1, show that SELENE achieves new SOTA performance. This demonstrates SELENE's suitability for homophilous networks and proves the flexibility of our designs. For the *Synthetic networks*, we present SELENE vs SDCN (SOTA node clustering model) vs GBT (SOTA graph contrastive learning model) vs FAGCN* (SOTA heterophilous GNN model) clustering accuracy in Figure 5-(a). SELENE achieves the best performance on the synthetic datasets, which shows SELENE adapts well to homophily/heterophily scenarios with(out) contextually raw node attributes. Moreover, FAGCN* performs worse on 10 synthetic networks, which indicates that the

heterophilous GNN models designed for supervised settings do not adapt well to unsupervised settings because they need supervision information to train the more complex aggregation mechanism. In addition, we observe that SELENE has no significant improvement with $h \leq 0.4$ because the network structure encoder's expressive power on extreme homophily is limited. Overall, SELENE still works significantly better for homophilous graphs, but deteriorates more gracefully compared to prior works while $h \to 0$.

**Node classification and link prediction.** Meanwhile, we acknowledge that some works adopt a different experiment pipeline that utilises node labels to train a classifier, after obtaining unsupervised node representations, to predict labels of test nodes (Velickovic et al., 2019; Peng et al., 2020; Bielak et al., 2021). Hence, we provide results of a number of real-world datasets that follow this setting to show the generality of SELENE. Results in Table 2 show that SELENE achieves the best classification performance on all datasets with homophily and heterophily. Moreover, we also report another major graph analysis task, i.e., link prediction, performances of a number of real-world datasets in Table 3. From the experimental results, we can find that *(i)* the homophily ratio has a significant influence on the link prediction task, and *(ii)* SELENE performs priority on this task as well. For more experimental settings and experimental results discussion, refer to Appendix D.

Table 2: Node classification results on real-world datasets. The **bold** and underline numbers represent the top-2 results.

| Dataset | F1-score | AE | N2V | S2V | GAE | SAGE | SDCN | DGI | GMI | GBT | H2GCN* | FAGCN* | GPRGNN* | Ours | ↑ (%) |
|---|---|---|---|---|---|---|---|---|---|---|---|---|---|---|---|
| | | | | | | Heterophilous datasets | | | | | | | | | |
| Texas | Micro | 0.598 | 0.509 | 0.592 | 0.5 | 0.563 | 0.552 | 0.541 | 0.505 | 0.461 | 0.523 | 0.607 | 0.561 | **0.643** | 5.93 |
| | Macro | 0.284 | 0.165 | 0.335 | 0.253 | 0.301 | 0.153 | 0.266 | 0.295 | 0.253 | 0.186 | 0.265 | 0.262 | **0.381** | 13.73 |
| Actor | Micro | 0.284 | 0.240 | 0.227 | 0.254 | 0.277 | 0.252 | 0.272 | 0.272 | 0.245 | 0.244 | 0.281 | 0.284 | **0.341** | 20.07 |
| | Macro | 0.194 | 0.178 | 0.190 | 0.164 | 0.204 | 0.125 | 0.244 | 0.195 | 0.224 | 0.138 | 0.166 | 0.192 | **0.274** | 12.30 |
| USA-Air. | Micro | 0.541 | 0.239 | 0.254 | 0.309 | 0.311 | 0.302 | 0.338 | 0.253 | 0.35 | 0.535 | 0.527 | 0.453 | **0.565** | 4.44 |
| | Macro | 0.476 | 0.235 | 0.253 | 0.236 | 0.299 | 0.288 | 0.260 | 0.221 | 0.288 | 0.483 | 0.454 | 0.370 | **0.532** | 10.14 |
| Bra.-Air. | Micro | 0.611 | 0.247 | 0.301 | 0.331 | 0.312 | 0.304 | 0.321 | 0.329 | 0.371 | 0.465 | 0.507 | 0.467 | **0.715** | 17.02 |
| | Macro | 0.538 | 0.203 | 0.284 | 0.244 | 0.295 | 0.281 | 0.244 | 0.249 | 0.319 | 0.415 | 0.420 | 0.426 | **0.692** | 28.62 |
| | | | | | | Homophilous datasets | | | | | | | | | |
| DBLP | Micro | 0.746 | 0.269 | 0.334 | 0.768 | 0.732 | 0.638 | 0.792 | 0.811 | 0.744 | 0.413 | 0.560 | 0.643 | **0.813** | 0.25 |
| | Macro | 0.737 | 0.227 | 0.305 | 0.758 | 0.721 | 0.633 | 0.784 | 0.804 | 0.736 | 0.371 | 0.536 | 0.585 | **0.810** | 0.75 |

Table 3: Link prediction results on real-world datasets. The **bold** and underline numbers represent the top-2 results.

| Dataset | Metric | AE | N2V | S2V | GAE | SAGE | SDCN | DGI | GMI | GBT | H2GCN* | FAGCN* | GPRGNN* | Ours |
|---|---|---|---|---|---|---|---|---|---|---|---|---|---|---|
| | | | | | | Heterophilous datasets | | | | | | | | |
| Texas | ROC-AUC | 0.529 | 0.549 | 0.571 | 0.634 | 0.559 | 0.640 | **0.787** | 0.777 | 0.783 | 0.451 | 0.673 | 0.782 | 0.784 |
| Actor | ROC-AUC | 0.501 | 0.673 | 0.652 | 0.668 | 0.634 | 0.552 | 0.626 | 0.609 | 0.598 | 0.679 | 0.613 | **0.796** | 0.725 |
| USA-Air. | ROC-AUC | 0.518 | 0.714 | 0.685 | 0.842 | 0.595 | 0.528 | 0.857 | 0.854 | 0.867 | 0.667 | 0.642 | **0.925** | 0.871 |
| Bra.-Air. | ROC-AUC | 0.665 | 0.592 | 0.637 | 0.787 | 0.615 | 0.677 | 0.855 | 0.791 | 0.786 | 0.702 | 0.696 | 0.838 | **0.881** |
| | | | | | | Homophilous datasets | | | | | | | | |
| DBLP | ROC-AUC | 0.872 | 0.758 | 0.717 | 0.870 | 0.841 | 0.628 | **0.939** | 0.884 | 0.892 | 0.736 | 0.764 | 0.890 | 0.908 |

## 6.4 Analysis

**Model scalability.** The network structure encoder module is categorised as a local network algorithm (Teng, 2016), which only involves local exploration of the network structure. On the other hand, the node attribute encoder module naturally supports the mini-batch mechanism. Therefore, our design enables SELENE to scale to representation learning on large-scale networks and to be friendly to distributed computing settings (Qiu et al., 2020). Table 1 illustrates that three competing methods, i.e., struc2vec, GMI and GBT, have out-of-memory issues on large datasets, i.e., Pubmed and Deezer., an issue which did not arise with SELENE. This shows SELENE's advantage in handling large-scale networks due to its local network algorithm characteristic.

**Effectiveness of loss function.** The objective loss function of SELENE contains three components, and we thus sought to test the effectiveness of each component. In particular, we ablate each component and evaluate

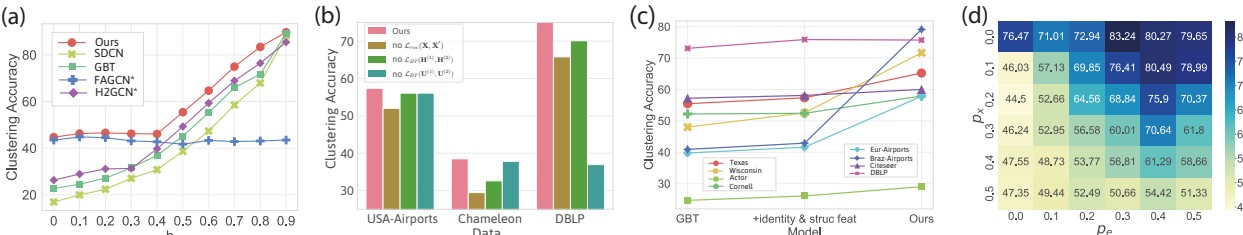

Figure 5: (a) Clustering accuracy comparison on synthetic networks. (b) Loss function ablation on real-world networks. (c) Framework component exploration on real-world networks. (d) Hyperparameter influences exploration on synthetic-0.8.

the obtained node representations on two heterophilous datasets (USA-Air., Chamel.) and one homophilous dataset (DBLP). Results shown in Figure 5-(b) indicate that the ablation of any component decreases the model's performance. Specifically, the ablation of $\mathcal{L}_{Rec}(\mathbf{X}, \widehat{\mathbf{X}})$ causes steeper performance degradation in heterophilous datasets, and the ablation of $\mathcal{L}_{BT}(\mathbf{U}^{(1)}, \mathbf{U}^{(2)})$ causes steeper performance degradation in homophilous datasets, which indicates the importance of node attributes and network structure information for heterophilous and homophilous networks, respectively (consistent with observations in Section 4).

**Effectiveness of dual-channel feature embedding pipeline.** SELENE contains a novel dual-channel features embedding pipeline to integrate node attributes and network structure information, thus, we conduct an ablation study to explore the effectiveness of this pipeline. We first remove the pipeline and only use the Barlow-Twins loss function to train a vanilla GCN encoding module (such a structure is the same as GBT, hence we remark it as GBT). Next, we add the network structure channel, which includes $r$-ego network extraction, anonymisation and distortion. Lastly, we add the node attribute channel to form the complete SELENE framework. Experimental results are shown in Figure 5-(c). Overall, we observe that the design of each channel is useful for learning better representation, with the node attribute channel playing a major role in the embedding of heterophilous networks. Note that adding the node attribute channel slightly decreases the clustering accuracy for the homophilous dataset, i.e., DBLP, but it is still competitive.

**Influence of $p_x$ and $p_e$.** We present SELENE's clustering accuracy on synthetic-0.8 with different $p_x$ and $p_e$ in Figure 5-(d). The figure indicates that hyperparameters of distortion methods significantly influence representation quality.

**Effectiveness of heterophily adopted self-supervised learning.** We intuitively discussed the importance of selecting a proper objective function for heterophily NE tasks in Section 5.3 and explained the reason for selecting reconstruction loss (Eq. 5) and BT loss (Eq. 4). Experimental results demonstrate the effectiveness of heterophily adopted self-supervised learning objective functions. For instance, SELENE shows priority compared with the network reconstruction loss-based model (GAE), distribution approximating loss-based model (N2V), node distance approximating loss-based model (SAGE), negative-sampling-based model (DGI) and mutual information-based model (GMI).

## 7 Conclusion and Future Directions

In this paper, we focused on the unsupervised network embedding task with challenging heterophily settings and tackled two main research questions. First, we showed through an empirical investigation that the performance of existing embedding methods that utilise network structure decreases significantly with the decrease of network homophily ratio. Second, to address the identified limitations, we proposed SELENE, which effectively fuses node attributes and network structure information without additional supervision. Comprehensive experiments demonstrated the significant performance of SELENE, and additional ablation analysis confirms the effectiveness of components of SELENE on real-world and synthetic networks. As future work, *(i)* we propose to explore the *Information Bottleneck* (Tishby & Zaslavsky, 2015; Wu et al., 2020) of network embedding to theoretically define the optimal representation of an arbitrary network (Zügner et al., 2018); *(ii)* designing a more powerful unsupervised network structure encoder for extreme heterophilous networks is also a promising future work.

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
