# OpenReview forum: "Unsupervised Network Embedding Beyond Homophily"
_TMLR — Accepted by TMLR_

### Review · Reviewer_b2WU · 2022-10-19

**Summary Of Contributions:**

The paper studies an interesting and timely problem, i.e., unsupervised network embedding on heterophilous graphs, while existing works mostly focus on either unsupervised network embedding with a homophily assumption or (semi)-supervised network embedding with heterophily. The proposed approach has multiple components, such as a dual-channel feature embedding pipeline, r-ego network feature extraction, and negative-sample free self-supervised learning objective. Extensive experiments demonstrate the performance improvements, and some ablation study is provided for further analysis.

**Audience:**

Yes

**Claims And Evidence:**

Yes

**Requested Changes:**

1. It is better to have a detailed discussion of the challenges in the studied problem and why existing approaches (or combinations) can not solve the problem.

2. It is suggested to thoroughly discuss the novelty, contributions, connections, and differences as compared with existing works.





**Strengths And Weaknesses:**

Strength:
1. The paper studies an interesting problem that is largely unexplored before.

2. The paper is clearly written and easy to follow with a clear logic.

3. The proposed framework is mostly reasonable and valid.

4. Extensive experiments on multiple datasets show significant performance improvements.


Weaknesses:
1. Although the paper studies a new setting (i.e., unsupervised network embedding with heterophily), the challenges of the problem are not clear. There is no detailed discussion about whether simple modification of existing approaches can not solve the problem. For instance, some simple combinations of GNN models with heterophily design trained by self-supervised learning loss as adopted in the paper.

2. Overall, the novelty of the proposed approach is a bit limited. It is more like an incremental combination of existing techniques proposed in multiple papers. In fact, almost all of the techniques are proposed and adopted in existing works. Therefore, the contribution of this work seems a bit weak.

3. Some claims in the paper might not be accurate. For instance, it is claimed that in the dual-channel feature embedding pipeline, node feature and network structure information are well utilized without interfering with each other. However, the node attributes are used as the input of the GNN model in the network structure encoder module. Therefore, at least the node attribute information will interfer the structure feature.

4. It is unclear why the double-channel feature embedding approach can solve the heterophily problem since the GNN part still aggregates information following the homophily assumption. It is suggested to discuss and explain why the node attribute separation will be better than existing works where residual connections of features are commonly used such as APPNP, GCNII, H2GCN and etc.


Other minor comments:
1. In section 4, how are the (semi)-supervised baselines trained under the unsupervised setting?

---

> ### Author Response · Authors · 2022-11-05
> **Response to Reviewer b2WU**
>
> > Although the paper studies a new setting (i.e., unsupervised network embedding with heterophily), the challenges of the problem are not clear. There is no detailed discussion about whether simple modification of existing approaches can not solve the problem. For instance, some simple combinations of GNN models with heterophily design trained by self-supervised learning loss as adopted in the paper.
>
> Thank you for your detailed comments. This paper involves several state-of-the-art GNN models with heterophily designs, such as H2GCN[1], FAGCN[2] and GPRGNN[3]. These models are trained with the same self-supervised learning loss adopted in the paper (without the dual-channel embedding pipeline). We rename the unsupervised adaptations of H2GCN, FAGCN and GPRGNN as H2GCN∗, FAGCN∗ and GPRGNN∗, due to the special training mechanism.
> The results, shown in Figure 3, Table 1,  Figure 5-(a), Table 2 and Table 3, demonstrate that combining heterophily GNN models with self-supervised learning loss does not work. More detailed discussions refer to Section 6.4.
>
> > Overall, the novelty of the proposed approach is a bit limited. It is more like an incremental combination of existing techniques proposed in multiple papers. In fact, almost all of the techniques are proposed and adopted in existing works. Therefore, the contribution of this work seems a bit weak.
>
> Please kindly note that TMLR evaluation criteria do not require papers to be extremely novel but only technically correct. Even a small modification to existing algorithms can be valuable. And the results of this paper would be interesting for many researchers of the TMLR community.
> We provide the first systematic empirical study of the influence of homophily ratio on network embedding tasks, and our results reveal the challenges of this task to the community. In addition, the proposed model shows signification priority in different scenarios, which provides a solution.
>
> > Some claims in the paper might not be accurate. For instance, it is claimed that in the dual-channel feature embedding pipeline, node feature and network structure information are well utilized without interfering with each other. However, the node attributes are used as the input of the GNN model in the network structure encoder module. Therefore, at least the node attribute information will interfer the structure feature.
>
> Given an attributed graph, we split $r$-ego network of each node into ego node attribute and network structure (Section 5.1). The node features that will be fed into the GNN encoder are another feature matrix, which contains manually designed node identity features and node structural features. They are different from $\mathbf{X}_{v}$.
>
> > It is unclear why the double-channel feature embedding approach can solve the heterophily problem since the GNN part still aggregates information following the homophily assumption. It is suggested to discuss and explain why the node attribute separation will be better than existing works where residual connections of features are commonly used such as APPNP, GCNII, H2GCN and etc.
>
> Given an attributed graph without any node class labels, there intuitively are two types of information to distinguish nodes, i.e., node attributes and the corresponding ego-network structure of each node. We argue a proper network embedding model should be able to distinguish nodes of different groups based on one or both resources. So we design the dual-channel embedding pipeline. Confusion information from one channel will not destroy another channel. Besides, our encoder modules and objective functions do not follow proximity-oriented and structure-oriented concepts. We encourage the model to embed each node’s original node attributes and ego network structure information into node representations. Like this, we argue the obtained node representations are well-distinguishable. And nodes of similar attributes will have similar node attribute representations. Similarly, nodes of similar ego-network structures will share similar structure context representations.
>
> [1] Beyond homophily in graph neural networks: Current limitations and effective designs. NeurIPS 2020.
>
> [2] Beyond low-frequency information in graph convolutional networks. AAAI 2021.
>
> [3] Adaptive universal generalized pagerank graph neural network, ICLR 2021.

---

### Review · Reviewer_2EgB · 2022-10-21

**Summary Of Contributions:**

This work studies unsupervised network embedding methods, where the embedding results can be used to identify node classes that could be either homophilic or heterophilic. The main idea is to not use the proximity-driven loss but to use self-supervised learning type of loss, Barlow-Twin loss on structural representations, and reconstruction loss on node features. Experiments show some superiority of the proposed method.

**Audience:**

Yes

**Broader Impact Concerns:**

No concerns on this.

**Claims And Evidence:**

No

**Requested Changes:**

1. The authors should clarify the details of their experiments on how their tune hyper-parameters and how to select the final clustering steps to address the weakness 3.

2. Experiments based on training a simple down-steam classifier should be done to verify the embedding quality to address the weakness 2.

3.Discuss the reference [1] above. And tune down the overclaim.

4. Check [2][3][4] above. And give proper explanation on the experiments over synthetic datasets. Compare with [2] and [3] in the experiments.

**Strengths And Weaknesses:**

Strengths:
1. There are very few works considering the setting of unsupervised learning that can handle node classification tasks over heterophilic networks. This is a very challenging problem. Although this work over-claims that they are the first one to consider this (explained later), research effort on this problem deserves appreciation.

2. The paper is generally written well. In particular, different parts of technical design get sufficiently motivated. In a high level, I think the way proposed to encode the network is intuitively good for node classification over heterophilic networks, while I think these are some technical issues which will be discussed later.

Weaknesses:
1. There are several points unclear to me. The paper says the experiments do not use any supervision signals in Appendix D. Then, my question is how the learned embeddings get mapped to node labels. Are there some pure unsupervised learning methods  adopted in the latent space such as k-means? If k-means-type of methods are used, have you tried different random seeds and how many time have you tried? We know that k-means is super sensitive to the initial seeds.  I believe this is super crucial to determine the performance of this method.

2. I do not quite buy the reasons in this paper that argue against previous works who train a simple down-steaming classifier over the embedding vectors to predict a small amount of labels. This is a valid method to measure the embedding quality as long as the classifier is simple and the number of used labels is small. I believe the proposed method should follow this protocol to do further comparison. Otherwise, how the adopted method maps embeddings to labels is super crucial, which is unclear to me. As I mentioned, k-means types of methods could be very sensitive to their initial seeds.

3. I do not quite believe the empirical results given that the authors claim that they are using purely unsupervised evaluation. The reason is that homophily and heterophily are two contradictory concepts. It is impossible that without any supervision, the learned embeddings are good for both simultaneously. Think about this. We have no knowledge about node labels and just compute network embedding. Also, we have no knowledge during the evaluation so we pick a kernel (entirely agnostic to node labels) to do final node clustering based on these embeddings. Okay, how can the proposed method simultaneously both cluster the nodes that are adjacent to each other in the network  (homophily) and also cluster the nodes that are not adjacent to each other in the network  (heterophily)? There must be a tradeoff.  Given this, I do not quite believe the current experiment results. Actually, a previous work [1], which also gets missed in the reference of this work, also considers unsupervised learning on heterophilic networks, where the results indeed show this tradeoff. I suggest the authors clarify the details of their experiments on how their tune hyper-parameters and how to select the final clustering steps. Also, I think experiments based on training a simple downsteaming classifier should be done to verify the embedding quality.

4. This is not the first paper on unsupervised learning for attributed heterogeneous network embedding. Please check [1] and give sufficient discussion.

5. The results on CSBM synthetic datasets also look weird because on these datasets heterophily is not an issue for GNNs, as one just needs to do substraction instead of addition when combining neighbor representations. Check arguments in empirical works [2][3] (consider using them as baselines also). Also, the theory in [4] on CSBM shows that GNN models with proper parameterization have no performance difference on homophilic and heterphilic CSBM.

6. What are unsupervised variants of H2GCN and FAGCN? I think the unsupervised loss function used here is crucial to determine their performances. These are important details.

Overall, I like the adopted technique and the paper writing. However, the experiments are unclear and contradictory. Also, there are a few overclaims and missing references. Once these weaknesses get addressed, I think this could be a good paper.

[1] Graph Auto-Encoder via Neighborhood Wasserstein Reconstruction, Tang et al. ICLR 2022
[2] Adaptive universal generalized pagerank graph neural network, Chien et al. ICLR 2021
[3] Is homophily a necessity for graph neural networks? Ma et al., ICLR 2022
[4] Understanding non-linearity in graph neural networks from the bayesian-inference perspective, Wei et al. NeurIPS 2022.

---

> ### Author Response · Authors · 2022-11-05
> **Response to Reviewer 2EgB (1)**
>
> > Then, my question is how the learned embeddings get mapped to node labels. Are there some pure unsupervised learning methods adopted in the latent space such as k-means? If k-means-type of methods are used, have you tried different random seeds and how many time have you tried? We know that k-means is super sensitive to the initial seeds. I believe this is super crucial to determine the performance of this method.
>
> Thank you for your insightful comments. We met this question as well in our experiments. We agree that evaluating the quality of embeddings is impossible if we do not consider any benchmark information. Our solution in this paper is that we only know the number of existing different class labels of each dataset, and we use this information to set up the \textit{K}--means cluster for the node clustering task.
> In addition, thank you for noticing the importance of random seeds. We forgot to report this part in the previous version. We added an implementation description in Appendix C. For the \textit{K}-means algorithm, we follow the setting of [5] that adopt the default Sklearn implementation. In addition, the random seed of different parts of our pipeline is related to experimental performance; thus, we design a $seed$ module in our code to set up fixed random seed for the relevant libraries, such as Numpy, Pytorch and Random. The final clustering section is thus repeated $10$ times with a set of seeds, and we report the mean performance.
>
> > I do not quite buy the reasons in this paper that argue against previous works who train a simple down-steaming classifier over the embedding vectors to predict a small amount of labels. This is a valid method to measure the embedding quality as long as the classifier is simple and the number of used labels is small. I believe the proposed method should follow this protocol to do further comparison. Otherwise, how the adopted method maps embeddings to labels is super crucial, which is unclear to me.
>
> We report the additional experimental results in Section 6.4, including node classification with a small number of labels (Table 2) and link prediction (Table 3).
> This paper mainly focuses on the very interesting node clustering task because many practical applications do not have valid or reliable class labels (as described in Section 1 and Section 2), such as many biomedical problems. And this problem is not new, a series of algorithms were studied, as discussed in Section 2, but not many researchers combine node clustering with heterophilous network problems. We hope our paper can reveal this interesting problem to the community.
>
> > It is impossible that without any supervision, the learned embeddings are good for both simultaneously. … Given this, I do not quite believe the current experiment results. Actually, a previous work [1], which also gets missed in the reference of this work, also considers unsupervised learning on heterophilic networks, where the results indeed show this tradeoff.
>
> Thank you for your constructive comments and for pointing out this nice paper [1] that we overlooked. We agree it is challenging to learn node representations work for both homophily and heterophily. Unlike [1], which wants to learn neighbourhood information, this paper considers this task differently.
> Given an attributed graph without any node class labels, there intuitively are two types of information to distinguish nodes, i.e., node attributes and the corresponding ego-network structure of each node. We argue a proper network embedding model should be able to distinguish nodes of different groups based on one or both resources. So we design the dual-channel embedding pipeline. Confusion information from one channel will not destroy another channel. Besides, our encoder modules and objective functions do not follow proximity-oriented and structure-oriented concepts. We encourage the model to embed each node’s original node attributes and ego network structure information into node representations. Like this, we argue the obtained node representations are well-distinguishable. And nodes of similar attributes will have similar node attribute representations. Similarly, nodes of similar ego-network structures will share similar structure context representations. The conclusion of [3] partially echoes our opinion that if we do not consider node attributes, nodes of the same group share similar neighbourhood patterns and different classes have distinguishable patterns, then we can have valid node representations.

---

> > ### Author Response · Authors · 2022-11-05
> > **Response to Reviewer 2EgB (2)**
> >
> > > This is not the first paper on unsupervised learning for attributed heterogeneous network embedding. Please check [1] and give sufficient discussion.
> >
> > We added discussions of [1] in Section 1 and Section 2 accordingly and revised our claims.
> >
> > > The authors should clarify the details of their experiments on how their tune hyper-parameters.
> >
> > We described the hyperparameter settings in Appendix C. We adopt available public implementations from the internet, and the comparison pipeline follows the settings of [5].
> >
> > > What are unsupervised variants of H2GCN and FAGCN? I think the unsupervised loss function used here is crucial to determine their performances. These are important details.
> >
> > In this paper, we train H2GCN, FAGCN and GPRGNN using the same mechanism as GBT to adapt them to the unsupervised setting, which is the same as the SELENE. We rename the unsupervised adaptations of H2GCN, FAGCN and GPRGNN as H2GCN$^*$, FAGCN$^*$ and GPRGNN$^*$, respectively.
> >
> > > The results on CSBM synthetic datasets also look weird because on these datasets heterophily is not an issue for GNNs, as one just needs to do substraction instead of addition when combining neighbor representations. Check arguments in empirical works [2][3] (consider using them as baselines also).
> >
> > [2] incorporates the learnable weights into representations of each layer via the Generalised PageRank technique. We added experimental results of [2] in Table 1, Table 2 and Table 3. As shown in the results, we find that its performance is influenced by strong heterophily when there are no training signals.
> > [3] utilises the vanilla GCN model to do a series of empirical studies to investigate what kind of “heterophily” influences GCN’s performance. They proposed Cross-Class Neighbourhood Similarity (CCNS) and introduced an additional parameter $\gamma$ to control CCNS. [3] found that higher $\gamma$ results in worse GCN performance. In our paper, we don’t have this additional parameter, i.e., $\gamma$, and our synthetic dataset can be treated as $\gamma = 0.5$. From Figure-3 or [3], we can observe a similar trend as our Figure 5-(a).
> > In our experiments, GCN is involved as a part of GAE, SDCN, DGI, and GBT baselines.
> >
> > > Also, the theory in [4] on CSBM shows that GNN models with proper parameterization have no performance difference on homophilic and heterphilic CSBM.
> >
> > We added discussions about [4] in Section 1, Section 2.
> >
> > [1] Graph Auto-Encoder via Neighborhood Wasserstein Reconstruction, Tang et al. ICLR 2022
> >
> > [2] Adaptive universal generalized PageRank graph neural network, Chien et al. ICLR 2021
> >
> > [3] Is homophily a necessity for graph neural networks? Ma et al., ICLR 2022
> >
> > [4] Understanding non-linearity in graph neural networks from the bayesian-inference perspective, Wei et al. NeurIPS 2022.
> >
> > [5] Structural deep clustering network. WWW 2020.

---

> > ### Comment · Reviewer_2EgB · 2022-11-13
> > **Thanks for the response (1)**
> >
> > Thanks for the response!
> >
> > I am still not convinced by the statement that one purely unsupervised node clustering method can do both well for homophilic and heterophilic networks. Because it is label-free, in principle, one can define homophilic labels (adjacent nodes more likely to share labels) or heterophilic labels (adjacent nodes less likely to share labels) arbitrarily. One clustering result cannot be good for both. This is nothing related to what encoding methods one adopts. Also, [1] encodes structural-oriented info, which actually shares a similar principle with the ego-network-structure used in this work. However, I do not want the discussion to be stuck at this point. As long as the authors make sure that the experiments were done rigorously and the current results are indeed what has been observed, I am okay with the statements.

---

> > > ### Author Response · Authors · 2022-11-17
> > > **Response to Reviewer 2EgB**
> > >
> > > Thank you for your insightful suggestions and this discussion.
> > >
> > > For the very interesting but often ignored process - node labelling. We think reviewer does not mean people can randomly label nodes in graphs but can choose either homophily or heterophily strategies to do node labelling. We agree node clustering results may vary if we change the labelling strategy. The synthetic datasets almost simulate what the reviewer described. Graph structure is similar, but node label distribution between neighbours is different across different datasets. And results of Figure 5-(a) tell us clustering accuracy decreases when h decreases from 0.9 to 0.5. But it keeps priority compared with other competing models.
> > >
> > > In addition, please note that SELENE encodes not only structure-oriented information but also very important node attributes. Our dual-channel feature embedding pipeline ensures information from both sides will not be neglected. This is inspired by the empirical study in Section 4 that pure node attributes provide significant information for node clustering tasks with heterophily. But this information will be invisible if we propagate it during message-passing, like other GNN models. Because of the lack of a strong training signal to train model parameters to identify it.
> > >
> > > About the experiments, we strictly follow the pipeline from benchmark work. We attached an anonymous implementation link. And there are already some works that utilise and compare with our public code on GitHub (with a different project name to satisfy the double-blind review process).

---

### Review · Reviewer_5GF4 · 2022-10-23

**Summary Of Contributions:**

The main contribution of this paper is SELENE, an unsupervised algorithm that learns low-dimensional node representations which can capture homophily but also heterophily in networks. The proposed method discriminates nodes' ego networks using node attributes and structural information separately. SELENE is evaluated in the tasks of node classification and link predicition. In the first task, it outperforms all baselines on all datasets, while in the second task, it outperforms them on most datasets.

**Audience:**

Yes

**Broader Impact Concerns:**

There are no concerns on the ethical implications of the work that would require adding a Broader Impact Statement.

**Claims And Evidence:**

Yes

**Requested Changes:**

Most requested changes are related to the weaknesses listed above.

- As discussed above, there is a lack of clear intuition regarding the proposed method. In my opinion, more intuition on why the method is designed as it is and why it works in heterophilous settings is necessary. I strongly encourage the authors to provide more intuition. Furthermore, I would suggest the authors provide a formal definition of heterophily and potentially some measure of heterophily (similar to equation 1, but for heterophily). I think that this would significantly strengthen the paper.

- The proposed method is mainly evaluated in the node classification task. I would expect methods that are evaluated in this task to belong to the family of supervised learning algorithms. I would thus suggest the authors whether experiment with a supervised variant of SELENE or evaluate the method in more downstream tasks. If no further downstream tasks exist, that would mean that unsupervised methods are not of any practical use.

- The comparison against some baselines is not fair since not all baselines can handle node attributes. For instance, node2vec cannot handle such attributes. I suggest the authors make sure that the list of baselines consists of methods that take the same data as the proposed model as input.

- It is well-known that $k$-means is sensitive to the initialization of the centroids. If I am not wrong, it is not reported in the paper how exactly the centroids were initialized. It is only mentioned that clustering was repeated 10 times. This is fairly important since different initialization schemes might lead to different results. Furthermore, it is mentioned in Appendix C that mean/std performance is reported, but I think only means are reported in the paper.

- In Section 6.4, the authors report the performance of the different methods in the node classification task, but they provide no details about the employed classifier and its hyperparameters. No details are provided in the Appendix as well. I suggest the authors provide those details since they are very important for reproducing the reported results.

- Typos:
page 2: "Out empirical study" --> "Our empirical study"

**Strengths And Weaknesses:**

Strengths
--
- The paper is well-written and easy to read, while it provides good background for the problem. The writing is very clear and pleasant to read. The authors also provide several examples that significantly contribute to the paper's clarity.

- The proposed method achieves state-of-the-art performance in the node classification and link prediction tasks. It outperforms all baselines on all node classification datasets and on 3 out of 5 link prediction datasets. Thus, it seems to be a good addition to the set of unsupervised node embedding algorithms.

Weaknesses
--
- While it is clear why methods that rely on the homophily assumption would fail in the heterophilous setting (because nodes in close proximity are mapped to similar vectors which is not necessarily the case in the heterophilous setting), it is not clearly explained in the paper why the proposed method can work well in such a setting. I suggest the authors go into more details such that the beneﬁts that the method brings are clear to the reader.

- This point is related to the previous one. In my view, heterophily is not properly defined in the paper. There are two different definitions of homophily, but no definition of heterophily. Would the following definition make sense: $heterophily = 1 - h$. I suggest the authors provide such a definition such that it becomes clear what are the properties of networks the proposed method aims to capture.

- Even though the paper focuses on the unsupervised heterophilous scenario, it is only evaluated in two downstream tasks, i.e., mainly in node classification and in link prediction. Since node classification is a supervised learning problem, and thus end-to-end models can be trained on the different datasets, I wonder whether the proposed method would be practically useful for such a task.

---

> ### Author Response · Authors · 2022-11-05
> **Response to Reviewer  5GF4 (1)**
>
> > As discussed above, there is a lack of clear intuition regarding the proposed method. In my opinion, more intuition on why the method is designed as it is and why it works in heterophilous settings is necessary.
>
> Thank you for your insightful comments. The design of SELENE is motivated by the observations from the empirical investigation (Section 4), we summarise three research challenges (RC) and propose three corresponding designs to address these three research challenges.
> For instance, motivated by the empirical observation in Section 4 that AE (only based on raw node features) has the only stable performance across h ∈ [0, 1) and graph structure-based NE approaches show priority when h → 1. Therefore, in Section 5.1, we have “RC1: How to leverage node attributes and network structure for NE?” We propose to deal with two important perspectives, i.e., node features and graph structure, separately.
> In Section 5.2, to answer “RC2: How to break the inherent homophily assumptions of traditional NE mechanisms?” We adopt the node attribute encoder module to preserve the original node attributes. We also introduce the network structure encoder to embed the structure-enhanced $r$-ego network of each node.
> In the end, we study “RC3: How to define an appropriate objective function to optimise the embedding learning process?” We first discuss the importance of choosing a proper objective function for network embedding with heterophily in Section 5.3. Specifically, existing NE objective functions inherently design positive and negative samples based on homophily assumption, which does not adapt to heterophily settings. Thus, we introduce the negative-sample-free self-supervised learning objectives, Reconstruction loss and BT loss, for framework optimisation.
>
> > I strongly encourage the authors to provide more intuition. Furthermore, I would suggest the authors provide a formal definition of heterophily and potentially some measure of heterophily (similar to equation 1, but for heterophily). I think that this would significantly strengthen the paper.
>
> Thank you for your suggestion. We give the formal definition of the heterophily ratio in Appendix D.
>
> > The proposed method is mainly evaluated in the node classification task. I would expect methods that are evaluated in this task to belong to the family of supervised learning algorithms. I would thus suggest the authors whether experiment with a supervised variant of SELENE or evaluate the method in more downstream tasks. If no further downstream tasks exist, that would mean that unsupervised methods are not of any practical use.
>
> Our proposed method supports node classification tasks, as discussed in Section 6.4. However, the node clustering task is the main downstream task we focus on. The major difference is that the node clustering models learn node representations without the guidance of supervision signals. We added discussions in Section 1 and Section 2 about the importance of node clustering tasks in different scenarios.
> Through the empirical study (Section 4), we found that existing network embedding models mostly fail with strong heterophily. In addition, we agree with the reviewer that if training node information is involved in the pipeline, the task will belong to the group of supervised learning algorithms, as we discussed in Appendix D. And our experimental settings for node clustering tasks can avoid this problem.
> In addition, we also studied another important graph analysis task, i.e., link prediction, and reported experimental results in Table 3. We can find that our proposed model is competitive on this downstream task as well.
>
> > The comparison against some baselines is not fair since not all baselines can handle node attributes. For instance, node2vec cannot handle such attributes. I suggest the authors make sure that the list of baselines consists of methods that take the same data as the proposed model as input.
>
> To provide a comprehensive empirical study to investigate the influence of homophily/heterophily on network embedding approaches, we adopt shallow and deep network embedding baseline models. There are 10 different baseline models that can utilise the same input as the proposed model.

---

> > ### Author Response · Authors · 2022-11-05
> > **Response to Reviewer 5GF4 (2)**
> >
> > > It is well-known that k-means is sensitive to the initialization of the centroids. If I am not wrong, it is not reported in the paper how exactly the centroids were initialized. It is only mentioned that clustering was repeated 10 times. This is fairly important since different initialization schemes might lead to different results.
> >
> > Thank you for your detailed comments. We added an implementation description in Appendix C. For the K-means algorithm, we follow the setting of [1] that adopt the default Sklearn implementation. In addition, the random seed of different parts of our pipeline is related to experimental performance, thus we design a $seed$ module in our code to set up fixed random seed for the relevant libraries, such as Numpy, Pytorch and Random.
> >
> > > Furthermore, it is mentioned in Appendix C that mean/std performance is reported, but I think only means are reported in the paper.
> >
> > Thank you for noticing this typo. We have revised it accordingly. We did not report std because we did not find a nice way to present so much numeric information in the table.
> >
> > > In Section 6.4, the authors report the performance of the different methods in the node classification task, but they provide no details about the employed classifier and its hyperparameters. No details are provided in the Appendix as well. I suggest the authors provide those details since they are very important for reproducing the reported results.
> >
> > The implementation description of additional experiments, including node classification and link prediction tasks, is described in Appendix D.
> >
> > > Typos: page 2: "Out empirical study" --> "Our empirical study"
> >
> > Thank you for noticing this typo. We have corrected it accordingly.
> >
> > [1] Structural deep clustering network. WWW 2020.

---

### Author Response · Authors · 2022-11-05
**Summary of changes**

Dear reviewers,
Thank you for taking the time to read our paper and provide us with valuable feedback. We made the changes according to your requests, which can be summarized as follows:

1. We added discussions on the importance of studying node clustering tasks throughout the paper.
2. We added the experimental results of a recommended baseline.
3. We added more experimental discussions to validate the effectiveness of our model design.
4. We introduce the neighbour sampler to avoid dense $r$-ego networks, which is described in Section 5.1 and Appendix C.

The changes can be seen in the uploaded version of the paper and supplementary material. We hope these modifications address your concerns, and we remain at your disposal.

---

### Decision · Action_Editors · 2022-12-04

**Recommendation:** Accept with minor revision

**Comment:**

This paper is well-written, with the ideas presented very clearly.  The proposed approach is simple and intuitive, and the results support that this approach works better than existing approaches.

On the other hand the novelty of the proposed approach is a bit limited, as it simply concatenates a node attribute embedding and a graph structural embedding, and uses distortions to encourage embeddings of the distorted versions to be similar rather than using proximity-based training objectives.  However, improvements in model performance do not always require significant novelty.  Simple ideas that really work could be even more useful.  So I don’t think this is necessarily a bad thing.

The paper still has some small issues, after the improvements the authors already made after feedback from the reviewers.  In particular:
1. Purely unsupervised clustering is fundamentally an ill-defined problem, without assumptions on the distribution of actual cluster labels nothing would work in the worst case.  The authors probably should make this point clear.  Even better, the paper could be revised to focus on this semi-supervised setting with a small amount of labels, which seems like a more natural setting for downstream tasks.
2. Related, reviewer 2EgB made a valid point that a purely unsupervised clustering method could not work simultaneously for graphs with homophilous and heterophilous labels, because they are fundamentally contradictory.  So it is worth discussing a bit further how to resolve this contradiction, probably (1) discuss the assumptions on the labeling distribution and (2) discuss that the proposed approach is not equally good for both homophilous and heterophilous graphs, it still works significantly better for homophilous graphs, but deteriorates more gracefully compared to prior works on heterophilous graphs.

**Audience:**

The paper addresses the problem of unsupervised representation learning for heterophilous graphs, all reviewers recognise that this is an important problem and the graph representation learning community should find this paper’s topic interesting.

Some of this paper’s settings may be a bit questionable, for example completely unsupervised clustering, which is fundamentally an ill-defined problem as the node labels could also be arbitrary.  Focusing more on the semi-supervised setting with a small set of labels could be a more realistic setting and would be of interest to more people.

**Claims And Evidence:**

This paper proposes a new method for unsupervised learning of network embeddings that does not rely on the homophily assumption, where near-by nodes are assumed to be similar.

The main claims of contribution in the paper include:
1. An empirical study of existing network embedding methods regarding how well they can handle graphs with homophily / heterophily.
2. The proposed approach SELENE is a good approach that doesn’t rely on the homophily assumption and outperforms / performs competitively relative to prior approaches.

Overall I think these claims are sufficiently supported.

For 1, the empirical study was mostly done in a controlled setting with synthetic data, but it is sufficiently illustrative.

For 2, the authors have presented comparison with a range of different prior approaches and the results support the promise of their proposed approach.

The paper previously made a few other claims that were revised later after the reviewer feedback, like being the first to handle unsupervised representation learning for attributed heterophilous graphs as pointed out by reviewer 2EgB.

---

> ### Author Response · Authors · 2022-12-19
> **Response to Action Editors**
>
> We appreciate the time and efforts you and the reviewers have dedicated to providing valuable feedback, affirmations, and insightful comments, which helped us improve the manuscript. Special thanks to the editor for the meticulous review process and organisation. We have been able to incorporate the changes to reflect the concerns provided by the editor for the camera-ready version.
>
> Since reviewers think our discussion about the experimental settings of node clustering and (semi-)supervised node classification is controversial, we carefully revised this part throughout the paper. We discuss the setting differences, but we do not argue their practical fairness (Appendix-D). Besides, we added explanations about the assumption of SELENE’s design (Section 1 and Section 5.4), that the design of SELENE assumes that nodes of different class labels share either similar node attributes or $r$-ego network structure. Under this assumption, we can learn node representations to distinguish nodes of different class labels from the perspective of node attributes or $r$-ego network structure. Moreover, we further highlight that even SELENE performs better than baselines on homophilous graphs, but SELENE has no significant improvement with $h \leq 0.4$ because the network structure encoder's expressive power on extreme homophily is limited. Overall, as the editor said, SELENE still works significantly better for homophilous graphs, but deteriorates more gracefully compared to prior works as $h \to 0$.

---

> > ### Comment · Action_Editors · 2022-12-22
> > **Reply**
> >
> > I found the sentence "the design of SELENE assumes that nodes of different class labels share either similar node attributes or r-ego network structure" a bit confusing, it's also not that clear from the context of your revised draft.
> >
> > If nodes of different class labels share these things how do you distinguish them from nodes of the same class?  It would be good to clarify this before approval.

---

> > > ### Author Response · Authors · 2022-12-22
> > > **Response to Action Editors**
> > >
> > > Thank you for pointing out this unclear part.
> > >
> > > We have revised it in the last paragraph of Section 5.4 and clarified the limitation of SELENE in terms of the capability to distinguish nodes of different classes.

---

> > > > ### Comment · Action_Editors · 2022-12-22
> > > > **Looks good**
> > > >
> > > > Thank you for the revision, this is much more clear now.